# Instability of Non-Standard Microsatellites in Relation to Prognosis in Metastatic Colorectal Cancer Patients

**DOI:** 10.3390/ijms21103532

**Published:** 2020-05-16

**Authors:** Francesca Pirini, Luigi Pasini, Gianluca Tedaldi, Emanuela Scarpi, Giorgia Marisi, Chiara Molinari, Daniele Calistri, Alessandro Passardi, Paola Ulivi

**Affiliations:** 1Biosciences Laboratory, Istituto Scientifico Romagnolo per lo Studio e la Cura dei Tumori (IRST) IRCCS, 47014 Meldola, Italy; francesca.pirini@irst.emr.it (F.P.); gianluca.tedaldi@irst.emr.it (G.T.); giorgia.marisi@irst.emr.it (G.M.); chiara.molinari@irst.emr.it (C.M.); daniele.calistri@irst.emr.it (D.C.); paola.ulivi@irst.emr.it (P.U.); 2Unit of Biostatistics and Clinical Trials, Istituto Scientifico Romagnolo per lo Studio e la Cura dei Tumori (IRST) IRCCS, 47014 Meldola, Italy; emanuela.scarpi@irst.emr.it; 3Department of Medical Oncology, Istituto Scientifico Romagnolo per lo Studio e la Cura dei Tumori (IRST) IRCCS, 47014 Meldola, Italy; alessandro.passardi@irst.emr.it

**Keywords:** mCRC, MSI, EMAST, combinatorial therapy, *VEGF-B*

## Abstract

Very few data are reported in the literature on the association between elevated microsatellite alterations at selected tetranucleotide repeats (EMAST) and prognosis in advanced colorectal cancer. Moreover, there is no information available in relation to the response to antiangiogenic treatment. We analyzed EMAST and vascular endothelial growth factor-B (VEGF-B) microsatellite status, together with standard microsatellite instability (MSI), in relation to prognosis in 141 patients with metastatic colorectal cancer (mCRC) treated with chemotherapy (CT) alone (*n* = 51) or chemotherapy with bevacizumab (B) (CT + B; *n* = 90). High MSI (MSI-H) was detected in 3% of patients and was associated with progression-free survival (PFS; *p* = 0.005) and overall survival (OS; *p* < 0.0001). A total of 8% of cases showed EMAST instability, which was associated with worse PFS (*p* = 0.0006) and OS (*p* < 0.0001) in patients treated with CT + B. A total of 24.2% of patients showed VEGF-B instability associated with poorer outcome in (*p* = 0.005) in the CT arm. In conclusion, our analysis indicated that EMAST instability is associated with worse prognosis, particularly evident in patients receiving CT + B.

## 1. Introduction

Although metastatic colorectal cancer (mCRC) is the third cause of cancer-related deaths worldwide [1,2], the last two decades have seen the introduction of new cytotoxic and biological agents that have improved treatment and overall survival (OS) [3], aided by a better understanding of the molecular mechanisms of the disease and the identification of new prognostic and predictive markers. The introduction of targeted therapies for mCRC, such as the anti-epidermal growth factor receptor (EGFR) monoclonal antibodies cetuximab and panitumumab, or the anti-vascular endothelial growth factor (VEGF-A) bevacizumab (B), has represented an important breakthrough in this setting, but there are still no validated predictive biomarkers for anti-angiogenic treatment [4].

Genomic instability is a landmark of mCRC and is the main effector that leads to the accumulation of mutations in repeated DNA sequences. Microsatellite instability (MSI) is a form of genomic instability caused by impairments in the mismatch repair (MMR) system [5]. MSI is characterized by loss of expression of MMR genes, typically consequent to either genetic mutation or epigenetic inactivation of the *MLH1* and *MSH2* gene promoters [6,7,8].

MSI occurs in around 15% of all sporadic colorectal cancers [9] and is associated with right-sided tumors, lower tumor stage at diagnosis, better prognosis, improved survival, and reduced recurrence of metastasis [10,11]. In the early stages of colorectal cancer, it has been demonstrated that MSI status plays a role in predicting response to adjuvant therapy. In particular, it has been observed that patients with high microsatellite instability (MSI-H) do not benefit from 5-fluorouracil (5-FU) adjuvant therapy [10,12,13], whereas a significant survival benefit has been reported from the administration of B after chemotherapy (CT) [14]. Similarly, in the QUASAR 2 randomized study, the addition of B to capecitabine led to improved survival with respect to capecitabine alone in MSI-H patients and in those with microsatellite stability (MSS) and low cluster of differentiation 31 (CD31) expression [15]. These data suggest that MSI-H status may be associated with a better response to anti-angiogenic drugs in an adjuvant setting. A study performed in the metastatic setting showed no difference in progression-free survival (PFS) after chemotherapy with bevacizumab therapy in relation to MSI status [16]. Conversely, a recent study performed on a large case series from the Cancer and Leukemia Group B (CALGB)/SWOG 80405 trial reported that patients with MSI-H benefited more from bevacizumab than from cetuximab, highlighting the need to further investigate the role of MSI in relation to the efficacy of B [17].

Elevated microsatellite alterations at selected tetranucleotide repeats (EMAST) are considered as a specific subtype of MSI and are caused by the defective translocation of MSH3 to the cytosol rather than by genetic or epigenetic alterations in MMR genes. EMAST are more frequent in colorectal cancers than MSI [18], and their presence is associated with advanced tumor stage, metastasis, poor survival, and intraepithelial inflammation [19,20,21,22]. The predictive role of EMAST has not been extensively investigated, probably because the repeat types and thresholds used in EMAST analysis have still not been standardized. However, some studies have reported a lack of correlation between EMAST and adjuvant therapy [23,24].

Although the canonical VEGF pathway promoting angiogenesis in mCRC involves the activation of the VEGF receptor 2 (VEGFR2) by VEGF-A, the concomitant stimulation of VEGFR1 signaling by VEGF-B may be specifically required to sustain metastasis by late-stage tumors, especially during anti-VEGF-A treatments [25]. VEGFR1 directly mediates mCRC cell migration and resistance to chemotherapy, indicating the existence of compensatory effects parallel to VEGF-A signaling [26]. Moreover, both VEGF-A and VEGF-B converge to VEGFR1 to induce tumor cell proliferation and survival and to promote the metastatic process [27]. However, despite the known importance of the *VEGF-B-K129Rfs*5* microsatellite frameshift deletion in mCRC [28], the association between *VEGF-B* gene microsatellite instabilities and clinical outcome in mCRC has still not been studied.

In the present study, we analyzed EMAST and *VEGF-B* status in relation to prognosis in a series of mCRC patients treated with CT alone or CT + B.

## 2. Results

### 2.1. Patient Characteristics

Clinical-pathological characteristics of patients involved in the study are reported in Table 1. Of the 141 cases analyzed, 90 were treated with CT + B and 51 with CT alone. Median age of the population was 68 years (range 34–85 years); 86 (61.0%) were male and 55 (39.0%) were female. A total of 109 (77.3%) patients had tumors located in the colon, whereas 32 (22.7%) had rectal tumors. Moreover, 75 (56.4%) and 58 (43.6%) were defined as left- and right-sided tumors, respectively. With regard to mutational status, 54 (38.3%) tumors had a KRAS Proto-Oncogene, GTPase (*KRAS*) mutation, 5 (11.1%) had a B-Raf Proto-Oncogene, Serine/Threonine Kinase (*BRAF*) mutation, and 2 (4.7%) had an NRAS Proto-Oncogene, GTPase (*NRAS*) mutation. All baseline patient characteristics were well balanced between CT alone and CT + B groups.

### 2.2. Microsatellite Status and Its Relation with Outcome

Of the 141 patients analyzed, 4 (3%) cases showed MSI-H and all received a B-based treatment. All of the MSI-H tumors were located in the right colon. After a median follow-up of 64 months (range 1–100), MSI-H patients showed a significantly lower median PFS (5.5 months, 95% CI 4.7–8.6 vs. 10.8 months, 95% CI 9.3–12.2; *p* = 0.005) and OS (6.5 months, 95% CI 4.7–9.4 vs. 26.6 months, 95% CI 23.3–31.6; *p* < 0.0001) than those with MSS tumors (Figure 1A,B). Using a multivariable model and adjusting for patient characteristics (age, gender, Eastern Cooperative Oncology Group performance status (ECOG PS), tumor location, CT and CT + B), microsatellite status remained significantly associated with PFS and OS. The PFS hazard ratio (HR) for MSI-H patients was 3.18 (95% CI 1.09–9.25; *p* = 0.034) and the OS HR was 6.45 (95% CI 2.08–19.99; *p* = 0.001).

### 2.3. EMAST Frequency and Patient Outcomes

Of the 135 patients analyzed for EMAST, 11 (8%) showed instability, defined as ≥ 2 unstable loci of the five canonical EMAST considered in the study according to the literature [19]. EMAST were more frequent in females (73%) and in tumors located in the right ascending colon (82%). In the overall case series, the presence of EMAST instability was indicative of a reduced probability of survival. Patients with EMAST-unstable tumors showed a shorter, albeit not significantly, median PFS of 5 months (95% CI 0.9–9.0) vs. 11.4 months (95% CI 9.4–12.6) (*p* = 0.060), and a significantly shorter OS (6.1 months 95% CI 1.5–25.2 vs. 27.3 months 95% CI 23.4–33.1; *p* = 0.006). These results were more evident in patients treated with CT + B (Table 2) who had shorter median PFS and OS than EMAST-stable patients, with 5 months (95% CI 0.9–9.0) vs. 11.7 months (95% CI 9.4–12.6) (*p* = 0.0006) for PFS and 6.1 months (95% CI 0.9–25.2) vs. 29.1 months (95% CI 23.3–34.5) (*p* < 0.0001) for OS (Table 2 and Figure 2A,B). Conversely, this was not confirmed in the group treated with CT alone due to the lack of patients with unstable tumors (Table 2, Figure 2A,B).

Considering the five EMAST markers individually in the overall case series (Table 3), patients with the MYCL Proto-Oncogene, BHLH Transcription Factor (*MYCL1*) marker instability showed a significantly shorter median PFS (6.1 months, 95% CI 1.8–9.1 vs. 11.4 months, 95% CI 9.4–12.6; *p* = 0.007) and OS (11.5 months, 95% CI 2.8–34.1 vs. 28.6 months, 95% CI 23.2–34.5; *p* = 0.030), as did patients with D20S85 marker instability (7.3 months, 95% CI 2.9–13.7 vs. 10.8 months, 95% CI 9.2–12.2; *p* = 0.022) and OS (8.1 months, 95% CI 4.4–24.0 vs. 28.6 months, 95% CI 23.4–34.1; *p* < 0.0001). Patients with D8S321 marker instability showed a significantly shorter median OS (17.8 months, 95% CI 4.7–26.4 vs. 28.8 months, 95% CI 23.3–35.7; *p* = 0.029). In the CT + B group, D20S85 instability was confirmed to be associated with a worse outcome, with shorter median PFS (5.5 months, 95% CI 0.9–12.6 vs. 11.4 months, 95% CI 9.3–12.4; *p* = 0.015) and OS (6.5 months, 95% CI 0.9–15.1 vs. 30.2 months, 95% CI 23.4–35.7; *p* < 0.0001), in patients with unstable vs. stable tumors, respectively. MYCL1, D20S82, and D8S321 instability was not significantly associated with shorter PFS and OS in this group (Table 3). Conversely, in patients treated with CT alone, instability of the MYCL1 marker was associated with shorter median PFS (3.1 months, 95% CI 0.6–9.1 vs. 10.2 months, 95% CI 8.3–18.2; *p* = 0.002) and OS (11.5 months, 95% CI 0.6–36.7 vs. 28.6 months, 95% CI 20.4–48.7; *p* = 0.014), whereas D20S82 marker instability was associated with longer median PFS (71.6 months, 95% CI 9.8–88.6 vs. 9.1 months, 95% CI 6.5–13.7; *p* = 0.017). However, the number of CT-only patients with unstable tumors was very small.

Using a multivariable model and adjusting for patient characteristics, MYCL1 instability was confirmed to be associated with worse prognosis in the overall case series, with shorter PFS (HR 2.14, 95% CI 1.17–3.90; *p* = 0.013) and OS (HR 1.89, 95% CI 1.02–3.50; *p* = 0.043). D20S85 instability, on the other hand, was associated with shorter OS (HR 3.62, 95% CI 1.73–7.59; *p* = 0.0007).

The interaction test between treatment (CT alone or CT + B) and EMAST status was not significant for EMAST, even though unstable patients treated with B often had a reduction in PFS approaching significance (*p* = 0.067). However, when markers were considered separately, D20S82 was correlated with a significantly poorer PFS (*p* = 0.004) and OS (*p* = 0.030), and MYCL1 was correlated with and a significantly worse PFS (*p* = 0.029), suggesting that instability in these two markers was predictive of a negative response to treatment.

### 2.4. Amplification of VEGF-B Microsatellite Showed a Tendency to Associate with Decreased Survival in Response to Chemotherapy

We first analyzed the mutational recurrence of short exonic mononucleotide repeats (AAAAAAAAG) that leads to the *K129Rfs* frameshift mutation and truncation of the *VEGF-B* transcript. Although *K129Rfs*5* frameshift deletion is the most frequent mutation of *VEGF-B*, as reported by the cBioPortal for Cancer Genomics in colorectal cancer datasets (Available online: www.cbioportal.org), we only found one patient harboring this mutation (deletion of one base c.379delA). Checking the University of California Santa Cruz (UCSC) Genome Browser database (Available online: http://genome.cse.ucsc.edu), we found that the *VEGF-B* gene also possesses a poly-dinucleotide microsatellite (normally 19xAG; hg38_chr11:64,237,086-64,237,123) whose somatic instability has never been reported. Interestingly, this microsatellite region is localized at a predicted splice junction (GTAG) between exons 4 and 5, which could be useful to determine the isoform transcripts of *VEGF-B* in mCRCs with genomic instability. We observed some variability in the number of AG dinucleotides among our patients, with a median length of 13xAG somatic repeats per allele. A total of 29 (24.4%) of the analyzed patients (*n* = 119) showed a shift in the number of repeats (amplification or shortening) in the tumor genome with respect to the germline, thus indicating the presence of genomic instability in this poly-AG microsatellite (Table 4). We did not find a correlation between *VEGF-B*–poly-AG instability and the other genomic markers EMAST and MSI or mutations in driver genes.

Overall, we observed a longer median PFS in patients with amplified poly-AG compared to those with a stable genotype. In patients treated with CT alone, amplification of the poly-AG was associated with reduced PFS compared to those with shortened poly-AG (7.0 vs. 28.0 months, respectively; *p* = 0.005), with a non-statistically significant trend towards poorer OS (Table 4). It is possible that the presence of an extended microsatellite at this site in *VEGF-B* played a role in tumor progression, which could not be stopped when chemotherapy was administered alone. Conversely, inclusion of B showed a tendency to reverse the outcome by prolonging the lifespan of patients with amplified poly-AG stretch in terms of both PFS and OS (although the difference in survival was not significant).

## 3. Discussion

Our results revealed that microsatellite instability, in terms of both MSI and EMAST, was associated with worse prognosis in mCRC, especially in patients receiving B-based treatment. MSI-H tumors are mainly observed in female patients, frequently located in the right colon, and are associated with poor prognosis with deep tumor invasion and poor histologic differentiation [29]. With regard to response to CT in mCRC, some authors have reported a better response to a 5-FU-based treatment in MSI-H patients [30] and a better outcome in those treated with FOLFIRI regimen [31]. Results are less convincing regarding the response to oxaliplatin-based regimens [8,32], mainly due to the small number of MSI-H patients in the various studies, which limits the possibility of drawing solid conclusions. In our study, there were no MSI-H patients treated with CT alone. With regard to B treatment, although there is ample evidence of its efficacy in MSI-H patients treated in an adjuvant setting [33], there are fewer findings of its benefit in metastatic disease. In a retrospective study on 140 mCRC patients receiving CT + B, no differences in outcome were observed between patients with MSI-H tumors and MSS tumors [16]. Conversely, another study performed on patients enrolled in the CALGB/SWOG phase III trial reported improved survival in MSI-H patients treated with CT + B compared to CT with cetuximab. This difference was not observed in MSS patients [17]. In our study, we were not able to compare patient prognosis in the CT + B and CT alone groups, as no MSI-H patients were present in the latter group. However, the four MSI-H patients treated with CT + B showed a worse outcome compared to those with MSS tumors.

EMAST is a genetic signature found in colorectal cancers that is caused by somatic dysfunction of the MMR protein hMSH3 [22]. There are very few data in the literature about EMAST in relation to response to CT. A study performed in the adjuvant setting showed that stage II/III colorectal cancer patients with EMAST tumors responded equally as well as those with non-EMAST tumors [24]. There are no data on this in the metastatic setting. To the best of our knowledge this is the first study to evaluate the correlation between EMAST and response to CT or CT + B in mCRC. Our results showed that the presence of EMAST instability was associated with a worse prognosis in the overall case series, with a trend that seemed more significant in the group of patients treated with CT + B. As far as we know, there are no previous publications reporting similar data on the association between each specific EMAST marker and prognosis and response to mCRC treatments; only a few reports have measured the frequency of individual markers in mCRC patients. Two studies reported that, among EMAST markers, D20S8 was the locus with the highest frequency of frameshift alterations in mCRC [19,20], and that its instability was a direct consequence of hMSH3 deficiency in tumor cells [34]. This marker is located in the chromosome region 20p12.3, typically associated with cancer susceptibility [35,36,37]. Alterations in MYCL1 are strongly related to recurrence and survival, and it has been linked to poor prognosis in mCRC [38,39]. MYCL1 is a structurally complex microsatellite consisting of mono-, tetra-, and pentanucleotide repeats [40,41], but preferentially mutated in the tetranucleotide locus [42], and is frequently unstable in non-MSI-H. Kambara and colleagues hypothesized that the MYCL1 mutation, which is frequent in CRC cancer, may promote tumor growth through the regulation of other genes [38]. Thus, the involvement of these two EMAST markers in tumorigenesis and prognosis in CRC colon cancer is sustained by previous literature. In our case series, MYCL1 was significantly associated with worse prognosis in patients treated with CT, whereas D20S85 correlated with poor survival in those receiving CT + B. Instability of both MYCL1 and D20S85 was also strongly indicative of poor OS (Table 3).

The VEGF family of growth factors comprises several members, of which VEGF-A has received the most attention, given its pivotal role in driving tumor angiogenesis [43]. The action of VEGF-A can, however, converge with that of VEGF-B to stimulate the formation of blood vessels through the activation of the VEGFR1 receptor, especially during metastatic progression [27,44]. There is evidence to suggest that a compensatory mechanism of VEGF-B/VEGFR1 signaling is involved in therapy resistance and anti-VEGF-A treatment [25,26]. It has also been shown that mCRC patients with MSI-H have higher levels of serum VEGF-A [45], that MSI-H tumors have increased angiogenic capacity [46], and that the presence of MSI could influence the effect of anti-VEGF-A therapies [47]. We wanted to investigate whether the clinical effects of the inhibition of VEGF signaling via B could be attributed to the instability of the *VEGF-B* gene. In fact, alterations in VEGF-B function might be useful for evaluating clinical response in mCRC, as the somatic *VEGF-B-K129Rfs*5* frameshift mutation, albeit rare, is the most important mutation of *VEGF-B* in CRC [28]. This alteration leads to a nucleotide deletion because of polymerase slippage on a short poly-A repeat microsatellite, which causes premature truncation of the VEGF-B peptide [48]. In our case series, only 1 of the 145 cases analyzed was found to be mutated for this microsatellite. This may be attributable to the fairly low frequency of somatic mutations in *VEGF-B* observed in CRC (0.4%; www.cbioportal.org), but a bigger cohort is needed to identify any potential association with therapy. We also wanted to see whether, in the event of altered microsatellite stability, other microsatellite alterations of *VEGF-B* are capable of predicting clinical outcome. A poly-AG repeat was identified in intron 4 of *VEGF-B*. This poly-AG microsatellite is localized in the proximity of a potential splice junction between exons 4 and 5; its alteration could be responsible for truncating the isoform of VEGF-B. Interestingly, the *VEGF-B-K129Rfs*5* frameshift mutation, which leads to VEGF-B protein truncation, is localized in exon 5. We found that 24.8% of tumors showed instability of this microsatellite. Overall, the presence of amplified poly-AG microsatellite was associated with poorer PFS, and this association became more significant when patients were treated with CT alone. This observation is interesting, given that VEGFR1, the cognate receptor of VEGF-B, directly mediates chemo-resistance in mCRC cells [25,26]. Conversely, the addition of B to CT was beneficial for outcome by prolonging the survival of patients with amplified poly-AG. This effect may be attributable to the inhibitory action exerted by B during altered functionality of VEGF-B.

## 4. Materials and Methods

### 4.1. Case Series

This study was conducted on patients involved in the multicenter randomized phase III ITACa trial (EudraCT no. 2007-004539-44 and on ClinicalTrials.gov NCT01878422) who were randomized to receive first-line chemotherapy (CT) alone, consisting of FOLFIRI or folinic acid/fluorouracil/oxaliplatin(FOLFOX), or CT with bevacizumab (B), and also on consecutive patients taking part in an independent biological prospective study conducted at our institute (IRST IRCCS, Meldola, Italy) with the same eligibility criteria and procedure as that of the ITACa trial, in which patients received first-line CT with B. All subjects gave written informed consent to participate in the studies. Both studies were conducted in accordance with the with the principles laid down in the Declaration of Helsinki, and the protocols were approved in 2009 by the Local Ethics Committee (Comitato Etico Area Vasta Romagna e IRST; Protocol code: IRST 153 01). Overall, 141 patients with available biological material were considered, of whom 90 received CT with B and 51 received CT only. All patients were assessed for response to treatment, PFS and OS, as per RECIST (response evaluation criteria in solid tumors) criteria (version 1.1.) Tumor response was evaluated every 8 weeks by CT scan. Responders were defined as patients who obtained complete response (CR) and partial response (PR), whereas non-responders were defined as those who showed stable disease (SD) and progressive disease (PD).

### 4.2. Genomic DNA Extraction

Both peripheral blood samples and formalin-fixed paraffin-embedded (FFPE) tumor tissue samples were available for 141 patients. Genomic DNA (gDNA) was extracted from whole blood using a QIAamp DNA Minikit (QIAGEN, Milan, Italy) following the manufacturer’s protocol. DNA was extracted from FFPE tumor tissue, starting from 5 μM sections. Tissue was lysed in 50 mM of KCl, 10 mM of Tris–HCl (pH 8.0), 2.5 mM of MgCl_2_, and Tween-20 (0.45%), and supplemented with proteinase K at a concentration of 1.25 mg/mL to be incubated overnight at 56 °C. Proteinase K was inactivated at 95 °C for 10 min and samples were then centrifuged twice to eliminate debris. The supernatant was evaluated for DNA quality and quantity by Nanodrop (Celbio Spa, Milan, Italy), and then underwent molecular analysis.

### 4.3. Microsatellite Instability Assay

All samples were tested for MSI using the CC-MSI kit (AB Analitica, Padova, Italy), which considers 13 microsatellite indicators: the five Bethesda panel markers (BAT-25, BAT-26, D5S346, D17S250, D2S123), six other markers (BAT40, NR21, NR24, D18S58, TGFβRII, D18S58), and two controls (TPOX, TH01) [49]. MSI status of each CRC case was classified into one of the following categories: microsatellite-stable (MSS), when no unstable markers were found; MSI-L, when 1-3 markers were determined as unstable; and MSI-H, when > 4 markers were found to be unstable.

### 4.4. Elevate Microsatellite Alterations at Selected Tetranucleotide Repeats (EMAST) Assay

Five tetranucleotide markers (MYCL1, D20S85, D8S321, D20S82, D9S242) were chosen from the literature and used to define EMAST status [19,24]. Tumor tissue and whole blood gDNA was amplified by PCR with specific labeled primers (Thermo Fisher Scientific, Waltham, MA, USA) using Kapa2G Robust Hotstart PCR Kit (Kapa Biosystems Roche, Wilmington, MA, USA) or Takara ExTaq (Diatech Pharmacogenetics, Jesi, Italy) at the conditions reported in Appendix A.

Fluorescent labeled fragments were analyzed by the Applied Biosystems 3130 (four-capillary) Genetic Analyzer (Applied Biosystems, Foster City, CA, USA), and the presence of frameshift mutations at tetranucleotide repeats was inferred by comparing the electropherograms of tumor versus blood. A locus was considered unstable if there was a frameshift difference (+/− multiples of four nucleotides) in the number of repeats between tumor and normal samples. We classified samples as EMAST-positive whey they showed ≥ 2 unstable markers compared with the blood.

### 4.5. VEGF-B Frameshift Mutation and Microsatellite Instability

To detect the presence of the somatic frameshift deletion of the *VEGF-B* mononucleotide repeat AAAAAAAAG (*K129Rfs*), located in exon 5, and the instability of the intronic poly-AG microsatellite (19xAG), we designed a pair of PCR primers spanning the *VEGF-B* genomic region chr:chr11:64,236,960-64,237,380 from intron 4 to exon 5 (forward primer: TGGGCAAGAAGAGGGAAACA; left primer: GGTGGGAGGAGAAAGAGGAG); for the PCR reaction we used 150 ng of gDNA extracted from FFPE tumor tissue, and 100 ng of gDNA extracted from the corresponding whole blood sample, as a control of germline variation. Amplified gDNA was column-purified with QIAquick PCR purification kit (QIAGEN, Hilden, Germany), and 1.5 µL of purified gDNA was subjected to Sanger sequencing using the BigDye Terminator v3.1 Cycle Sequencing Kit (Applied Biosystems, Foster City, CA, USA). Labeled DNA was cleaned by DyeEx 2.0 Spin Kit (QIAGEN, Hilden, Germany) to remove unincorporated dye terminators from the sequencing reaction, before proceeding with capillary gel electrophoresis (Applied Biosystems 3130 Genetic Analyzer) and DNA analysis. We defined the 19xAG repeat as unstable when the number of the microsatellite duplets in the tumor DNA was either higher (amplified AG microsatellite) or lower (shortened AG microsatellite) than that of the gDNA, taking into account both *VEGF-B* alleles separately.

### 4.6. Statistical Analysis

The purpose of the present study was to examine the association between MSI, EMAST, and PFS and OS in mCRC patients treated with bevacizumab-based CT or CT alone. PFS was the primary aim and overall response rate (ORR) and OS were secondary efficacy endpoints. PFS was calculated as the time elapsed between the date of randomization/registration and the date of the first documented evidence of disease progression or last tumor evaluation or death in the absence of progressive disease. OS was calculated as the time elapsed between the date of randomization/registration and the date of death from any cause or last follow-up visit.

The Kaplan–Meier method was used to describe time-to-event data (PFS, OS), and the log rank test was used to compare survival curves. Ninety-five percent confidence intervals (95% CI) were calculated using Greenwood’s formula. The Cox regression model was used to estimate hazard ratios (HR) and their 95% CIs. HRs were adjusted by center and baseline characteristics (gender, age, performance status, *KRAS* status, tumor localization (rectum/colon), stage at diagnosis, and chemotherapy regimen (FOLFOX4/FOLFIRI)). The selection of covariates was based on a number of suspected prognostic factors derived from the ITACa study.

The effect of the interaction between EMAST and treatment (CT + B or CT only) on PFS/OS was evaluated using Cox regression models of the entire population (CT + B and CT-only arms) that included EMAST, treatment, and treatment-by-EMAST status.

All *p*-values were based on two-sided testing, and statistical analyses were carried out using SAS statistical software version 9.4 (SAS Institute, Cary, NC, USA).

## 5. Conclusions

Our results showed that MSI, EMAST, and VEGF-B microsatellite instability was associated with prognosis in CRC patients treated with CT alone or CT + B. In particular, MSI and EMAST instability was associated with a worse prognosis in CT + B patients, whereas *VEGF-B* microsatellite instability was linked to poorer prognosis in those treated with CT alone. Although our findings warrant further investigation in an mCRC setting, we hypothesize that these markers may have some relevance for the clinical outcome of the disease.

## Figures and Tables

**Figure 1 ijms-21-03532-f001:**
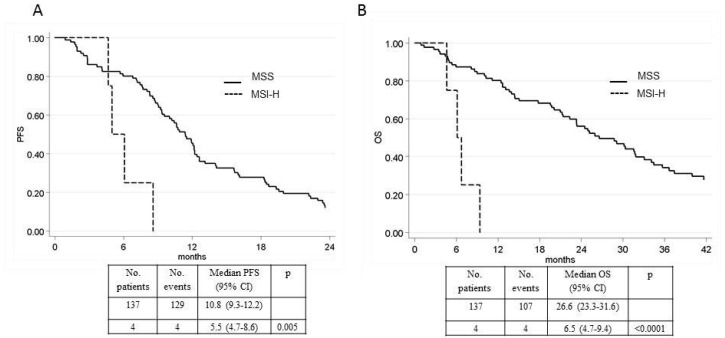
(**A**) Progression-free survival (PFS) in high microsatellite instability (MSI-H) and microsatellite stability (MSS) patients. (**B**) Overall survival (OS) in MSI-H and compared to MSS patients.

**Figure 2 ijms-21-03532-f002:**
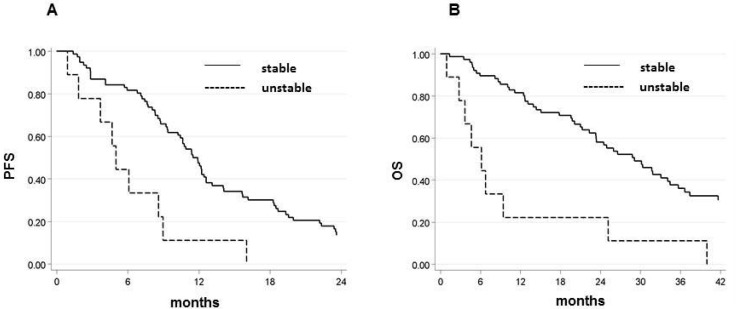
(**A**) PFS in EMAST-unstable and -stable patients in CT + B arm. (**B**) OS in EMAST-unstable and -stable patients in CT + B arm.

**Table 1 ijms-21-03532-t001:** Patient characteristics.

	Overall (*n* = 141)	CT + B Arm (*n* = 90)	CT Arm (*n* = 51)
	No. (%)	No. (%)	No. (%)
**Median age, years (range)**	68 (34–85)	69 (34–85)	66 (37–82)
**Gender**			
Male	86 (61.0)	57 (63.3)	29 (56.9)
Female	55 (39.0)	33 (36.7)	22 (43.1)
**ECOG PS**			
0	113 (81.3)	70 (79.6)	43 (84.3)
1–2	26 (18.7)	18 (20.4)	8 (15.7)
Unknown/missing	2	2	0
**Tumor location**			
Colon	109 (77.3)	69 (76.7)	40 (78.4)
Rectum	32 (22.7)	21 (23.3)	11 (21.6)
Left-sided	75 (56.4)	47 (56.6)	28 (56.0)
Right-sided	58 (43.6)	36 (43.4)	22 (44.0)
Unknown/missing	8	7	1
**Stage at diagnosis**			
I-III	20 (14.2)	8 (8.9)	12 (23.5)
IV	121 (85.8)	82 (91.1)	39 (76.5)
**CT regimen**			
Oxaliplatin-based	89 (63.1)	59 (65.6)	30 (58.8)
FOLFIRI	52 (36.9)	31 (34.4)	21 (41.2)
***KRAS***			
Wild type	87 (61.7)	52 (57.8)	35 (68.6)
Mutated	54 (38.3)	38 (42.2)	16 (31.4)
***NRAS***			
Wild type	41 (95.4)	29 (93.6)	12 (100)
Mutated	2 (4.7)	2 (6.5)	-
***BRAF***			
Wild type	40 (88.9%)	29 (90.6%)	11 (84.6%)
Mutated	5 (11.1%)	3 (9.4%)	2 (15.4%)

CT, chemotherapy; B, bevacizumab; ECOG PS, Eastern Cooperative Oncology Group Performance Status; FOLFIRI, Irinotecan/5-Fluorouracil/leucovorin; *KRAS*, KRAS Proto-Oncogene, GTPase; *NRAS*, NRAS Proto-Oncogene, GTPase; *BRAF*, B-Raf Proto-Oncogene, Serine/Threonine Kinase.

**Table 2 ijms-21-03532-t002:** Univariate analysis of PFS and OS for EMAST instability in the three patient groups.

	No. Patients	No. Events	Median PFS (Months) (95% CI)	*p*	No. Events	Median OS (Months) (95% CI)	*p*
**Overall**							
Stable	124	117	11.4 (9.4–12.6)		95	27.3 (23.4–33.1)	
Unstable	11	10	5.0 (0.9–9.0)	0.060	10	6.1 (1.5–25.2)	0.006
**CT + B**							
Stable	76	71	11.7 (9.4–12.6)		57	29.1 (23.3–34.5))	
Unstable	9	9	5.0 (0.9–9.0)	0.0006	9	6.1 (0.9–25.2)	<0.0001
**CT**							
Stable	48	46	9.8 (8.3–14.2)		38	27.1 (20.8–36.7)	
Unstable	2	1	nr	0.486	1	nr	0.697

PFS, progression-free survival; OS, overall survival; EMAST, elevated microsatellite alterations at selected tetranucleotide repeats; CT, chemotherapy; B, bevacizumab; nr, not reached.

**Table 3 ijms-21-03532-t003:** Univariate analysis of PFS and OS for single EMAST markers in patients treated with CT + B (A) or CT alone (B).

	No. Patients	No. Events	Median PFS (Months) (95% CI)	*p*	No. Events	Median OS (Months) (95% CI)	*p*
**Overall**							
**D20S82**							
Stable	108	102	10.6 (9.0–12.2)		86	26.0 (23.2–30.2)	
Unstable	21	20	9.0 (3.7–11.5)	0.793	17	14.0 (6.1–40.0)	0.925
**D20S85**							
Stable	123	115	10.8 (9.2–12.2)		93	28.6 (23.4–34.1)	
Unstable	12	12	7.3 (2.9–13.7)	0.022	12	8.1 (4.4–24.0)	<0.0001
**D8S321**							
Stable	108	104	10.5 (9.0–12.1)		83	28.8 (23.3–35.7))	
Unstable	11	10	8.6 (4.7–16.0)	0.576	10	17.8 (4.7–26.4)	0.029
**D9S242**							
Stable	119	113	10.8 (9.3–12.4)		94	26.4 (23.3–31.7)	
Unstable	9	8	6.5 (0.7–97.4)	0.427	6	27.1 (1.5–nr)	0.680
**MYCL1**							
Stable	107	100	11.4 (9.4–12.6)		82	28.6 (23.2–34.5)	
Unstable	15	14	6.1 (1.8–9.1)	0.007	13	11.5 (2.8–34.1)	0.030
**CT + B**							
**D20S82**							
Stable	65	61	11.9 (9.4–12.6)		50	26.0 (21.3–34.1)	
Unstable	17	17	7.4 (2.5–10.8)	0.078	16	12.7 (4.7–33.1)	0.113
**D20S85**							
Stable	75	70	11.4 (9.3–12.4)		56	30.2 (23.4–35.7)	
Unstable	10	10	5.5 (0.9–12.6)	0.015	10	6.5 (0.9–15.1)	<0.0001
**D8S321**							
Stable	74	71	11.1 (9.0–12.2)		57	28.8 (21.3–34.5)	
Unstable	8	7	7.3 (1.8–16.0)	0.599	7	8.1 (2.8–25.2)	0.075
**D9S242**							
Stable	75	71	11.4 (9.2–12.4)		59	26.0 (21.0–31.9)	
Unstable	4	4	7.0 (4.7–19.3)	0.338	3	23.1 (4.7–40.0)	0.518
**MYCL1**							
Stable	67	63	11.5 (9.2–12.6)		52	28.8 (21.0–34.5)	
Unstable	10	9	7.3 (1.8–12.2)	0.224	8	18.0 (2.8–49.4)	0.408
**CT**							
**D20S82**							
Stable	43	41	9.1 (6.5–13.7)		36	26.4 (20.2–36.7)	
Unstable	4	3	71.6 (9.8–88.6)	0.017	1	nr	0.048
**D20S85**							
Stable	48	45	9.5 (8.3–15.0)		37	27.1 (20.4–36.7)	
Unstable	2	2	12.5 (11.4–13.7)	0.847	2	30.8 (24.0–37.6)	0.778
**D8S321**							
Stable	34	33	9.2 (6.5–14.2)		26	29.1 (20.4–48.7)	
Unstable	3	3	13.1 (6.2–20.1)	0.7550	3	26.4 (20.1–28.6)	0.329
**D9S242**							
Stable	44	42	9.8 (8.3-15.0)		35	27.3 (20.4–37.3)	
Unstable	5	4	6.5 (0.7–97.4)	0.232	3	27.1 (1.5–nr)	0.594
**MYCL1**							
Stable	40	37	10.2 (8.3–18.2)		30	28.6 (20.4–48.7)	
Unstable	5	5	3.1 (0.6–9.1)	0.002	5	11.5 (0.6–36.7)	0.014

PFS, progression-free survival; OS, overall survival; EMAST, elevated microsatellite alterations at selected tetranucleotide repeats; CT, chemotherapy; B, bevacizumab; nr, not reached; MYCL1, MYCL Proto-Oncogene, BHLH Transcription Factor.

**Table 4 ijms-21-03532-t004:** Univariate analysis of PFS and OS for vascular endothelial growth factor-B (VEGF-B) instability in the three patient groups.

	No. Patients	No. Events	Median PFS (Months) (95% CI)	*p*	No. Events	Median OS (Months) (95% CI)	*p*
**Overall**							
VEGF-B stable	88	82	10.0 (8.9–12.2)		69	23.3 (20.1–31.7)	
AG shortened	12	12	14.5 (2.3–77.9)		10	27.7 (14.0–79.5)	
AG amplified	17	17	12.2 (7.4–19.3)	0.050	12	36.7 (9.4–71.7)	0.406
**CT + B**							
VEGF-B stable	53	50	10.0 (7.7–12.2)		43	21.4 (13.7–31.9)	
AG shortened	3	3	8.7 (2.3–10.2)		3	26.0 (13.9–46.1)	
AG amplified	13	13	18.3 (8.6–22.2)	0.132	8	44.0 (9.4–71.7)	0.159
**CT**							
VEGF-B stable	35	32	10.2 (6.3–15.0)		26	26.5 (20.2–39.7)	
AG shortened	9	9	28.0 (2.0–88.6)		7	28.0 (4.3–nr)	
AG amplified	4	4	7.0 (0.6–9.1)	0.005	4	22.6 (0.6–36.7)	0.268

PFS, progression-free survival; OS, overall survival; EMAST, elevated microsatellite alterations at selected tetranucleotide repeats; VEGF-B, vascular endothelial growth factor-B; CT, chemotherapy; B, bevacizumab.

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
