# Peer review of "Instability of Non-Standard Microsatellites in Relation to Prognosis in Metastatic Colorectal Cancer Patients"

_ijms, 2020, doi:10.3390/ijms21103532_

Round 1

Reviewer 1 Report

This work is well written and structured and is innovative. It represents one of the few works in literature that report this topic. I believe that the results obtained and well discussed can open new fields of research both from a biological and clinical point of view in the pathology of colorectal cancer.

Author Response

We thank the reviewer for the kind comments and positive evaluation of our work

Reviewer 2 Report

There were multiple grammatical and syntax errors in the manuscript. It is strongly recommended that the manuscript is reviewed by an English editor prior to resubmission.

As an example here are some of the errors noted:

Despite metastatic colon cancer (mCRC) is the third leading cause of cancer-related deaths – This is grammatically incorrect

due also to the increased knowledge – ‘also’ can be removed in this phrase

PFS after chemotherapy – please define PFS for the reader as this is the first time the term has been introduced

Of the 135 patients analyzed for EMAST, 11 (8%) resulted unstable – should be rephrased as “resulted in unstable”

and in right ascending colon tumor localization (82%) was observed. – This phrasing does not make sense.

Considering the 5 EMAST markers singly, - The word ‘singly’ should not be used.

In the CT+B group, D20S85 instability confirmed to be associated with worse outcome – should be rephrased as “20S85 instability is confirmed to be”

A different behavior was observed in patients treated with CT only – What is this behavior? This is not the best use of the term.

MYCL1 instability confirmed to be associated – this should be rephrased to “MYCLA instability is confirmed to”

Whereas D20S85 instability is associated with shorter OS – Whereas should be not be used to start a sentence

which somatic instability has never been reported – this should be rephrased

inclusion of bevacizumab in the treatment showed a tendency to revert the outcome – What does revert mean here? It is not the best use of the term

setting a bigger benefit from the drug in MSI-H patients [33] is well established – This should be rephrased

an evident improvement in survival – the term evident should be removed or substituted for another word such as significant

a trend seemed more significant in the group – should be rephrased to “with a trend that seemed more”

colleagues hypothesized that MYCL1 mutation, which is frequent in colorectal cancer – should change this phrasing to “that the MYCL1 mutation”

In our case set, we did find only one patient mutated for this microsatellite out of 145 cases analysed – What do you mean by case set? Are you referring to a case series?

Interestingly, VEGF-B-K129Rfs*5 frameshift mutation – should be rephrased to “the VEGF-B-K129Rfs*5 frameshift mutation

Author Response

We thank the reviewer for the detailed revision of our manuscript. We have it checked for English editing and proofreading of grammar errors. 

We have corrected all the issues raised by the reviewer as follow:

Despite metastatic colon cancer (mCRC) is the third leading cause of cancer-related deaths – This is grammatically incorrect

LINE 33-34, corrected: “Although metastatic colon cancer (mCRC) is the third leading cause of cancer-related deaths worldwide“.

due also to the increased knowledge – ‘also’ can be removed in this phrase

LINE 35-37, corrected: “aided by a better understanding of the molecular mechanisms of the disease and the identification of new prognostic and predictive markers“.

PFS after chemotherapy – please define PFS for the reader as this is the first time the term has been introduced

LINE 60: We have corrected accordingly.

Of the 135 patients analyzed for EMAST, 11 (8%) resulted unstable – should be rephrased as “resulted in unstable”

LINE 114-116, rephrased: “Of the 135 patients analyzed for EMAST, 11 (8%) resulted unstable showed instability, as defined by considering as ≥ 2 unstable loci of the five canonical EMAST considered in the study according to the literature“.

and in right ascending colon tumor localization (82%) was observed. – This phrasing does not make sense.

LINE 116-117, corrected: “EMAST were more frequent in females (73%) and in tumors located in the right ascending colon“.

Considering the 5 EMAST markers singly, - The word ‘singly’ should not be used.

LINE 137, we have corrected as “individually “.

In the CT+B group, D20S85 instability confirmed to be associated with worse outcome – should be rephrased as “20S85 instability is confirmed to be”

LINE 145, corrected: “D20S85 instability was confirmed to be associated with a worse outcome“.

A different behavior was observed in patients treated with CT only – What is this behavior? This is not the best use of the term.

LINE 149, corrected: “Conversely, in patients treated with CT alone“.

MYCL1 instability confirmed to be associated – this should be rephrased to “MYCLA instability is confirmed to”

LINE 160-162, corrected: “MYCL1 instability was confirmed to be associated“.

Whereas D20S85 instability is associated with shorter OS – Whereas should be not be used to start a sentence

LINE 162-163, corrected: “D20S85 instability, on the other hand, was associated with shorter OS “.

which somatic instability has never been reported – this should be rephrased

LINE 180, we substituted which with “whose”.

inclusion of bevacizumab in the treatment showed a tendency to revert the outcome – What does revert mean here? It is not the best use of the term

LINE 200, we have substituted reverte with “reverse”.

setting a bigger benefit from the drug in MSI-H patients [33] is well established – This should be rephrased

LINE 215, rephrased: “although there is ample evidence of its efficacy in MSI-H patients treated in an adjuvant setting“.

an evident improvement in survival – the term evident should be removed or substituted for another word such as significant

LINE 219, corrected: “phase III trial reported improved survival in MSI-H patients“.

a trend seemed more significant in the group – should be rephrased to “with a trend that seemed more”

LINE 232, we modified the sentence as suggested.

colleagues hypothesized that MYCL1 mutation, which is frequent in colorectal cancer – should change this phrasing to “that the MYCL1 mutation”

LINE 244, we corrected as suggested.

In our case set, we did find only one patient mutated for this microsatellite out of 145 cases analysed – What do you mean by case set? Are you referring to a case series?

LINE 248, we substituted set with “series”.

Interestingly, VEGF-B-K129Rfs*5 frameshift mutation – should be rephrased to “the VEGF-B-K129Rfs*5 frameshift mutation”

LINE 275, we corrected as suggested.